# Comparison of Different Remote Sensing Methods for 3D Modeling of Small Rock Outcrops

**DOI:** 10.3390/s20061663

**Published:** 2020-03-17

**Authors:** Tomáš Mikita, Marie Balková, Aleš Bajer, Miloš Cibulka, Zdeněk Patočka

**Affiliations:** 1Department of Forest Management and Applied Geoinformatics, Faculty of Forestry and Wood Technology, Mendel University in Brno, 613 00 Brno, Czech Republic; milos.cibulka@mendelu.cz (M.C.); zdenek.patocka@mendelu.cz (Z.P.); 2Department of Geology and Pedology, Faculty of Forestry and Wood Technology, Mendel University in Brno, 613 00 Brno, Czech Republic; marie.balkova@mendelu.cz (M.B.); ales.bajer@mendelu.cz (A.B.)

**Keywords:** terrestrial laser scanning, ARCore, Structure from Motion, rock outcrops

## Abstract

This paper reviews the use of modern 3D image-based and Light Detection and Ranging (LiDAR) methods of surface reconstruction techniques for high fidelity surveys of small rock outcrops to highlight their potential within structural geology and landscape protection. LiDAR and Structure from Motion (SfM) software provide useful opportunities for rock outcrops mapping and 3D model creation. The accuracy of these surface reconstructions is crucial for quantitative structural analysis. However, these technologies require either a costly data acquisition device (Terrestrial LiDAR) or specialized image processing software (SfM). Recent developments in augmented reality and smartphone technologies, such as increased processing capacity and higher resolution of cameras, may offer a simple and inexpensive alternative for 3D surface reconstruction. Therefore, the aim of the paper is to show the possibilities of using smartphone applications for model creation and to determine their accuracy for rock outcrop mapping.

## 1. Introduction

Light Detection and Ranging (LiDAR) and Structure from Motion (SfM) software provide large amounts of digital data from which virtual outcrops can be created. The accuracy of these surface reconstructions is crucial for quantitative structural analysis [1]. The use of photogrammetry and laser scanning significantly increased the possibilities of documentation of geological structures and archaeological monuments.

In [2] the Structure from Motion approach and the Dense Image Matching (DIM) or Multi View Stereo (MVS) algorithms are characterized by five main steps, which require, as their input data, only the acquired images:Feature extraction and identification of the matching points between images through different algorithms, such as the Scale Invariant Feature Transform (SIFT) algorithm [3], or its modified version [4];Outliers filtering, exploiting the Random Sample Consensus (RANSAC) algorithm [5], and a robust estimation of the acquisition geometry through linear models that impose geometric constraints;Estimate of the 3D coordinates of the object, internal and external orientation and the relative statistical information concerning the accuracy of the calculation, through the Bundle-Adjustment technique [6];Dense surface reconstruction of the point cloud identified for the orientation, exploiting several algorithms, including the Exact, Smooth, Height-field and Fast methods, for the object surface description and its main discontinuities [7];Dense point cloud interpolation and texturing in order to obtain a photorealistic visualization.

The SfM method offers an economical and simple solution for many industries, including geological sciences. Close-range terrestrial photogrammetry and computer vision approach is suitable to obtain high-resolution spatial data suitable for modelling meso and micro-scale landforms [8]. SfM produces models in which a high-resolution photograph is fully integrated with the underlying 3-D surface and is therefore suitable for rock slopes mapping [9]. This method is particularly useful at steep dam abutments as an alternative to rope access and can provide necessary accuracy for detailed analyses [10]. In [11] the authors recommend wherever possible the use of remote sensing derived 3D models in combination with high resolution photographs of the rock cuts. In tunnels, SfM is used to characterize underground rocks [12], to monitor deformations, displacements and to analyze tunnel stability [13,14]. Furthermore, the photogrammetric SfM method has its place in archeology [15]. Stones and rocks can be also part of cultural heritage and the method of 3D modelling can improve the archeological research. In [16] carved stones modelling by SfM was tested to obtain photo-realistic 3D models of the stones. 

The Terrestrial Laser Scanning (TLS) provides a high-density point cloud, up to thousands of points per square meter, depending on the setting parameters. During a short period of time, large numbers of high-quality data can be obtained to characterize the object of interest. TLS has a wide range of applications, especially in architecture and construction [17], but it can also be exploited in geosciences. Examples include archeology or cultural heritage documentation [18,19,20,21,22,23], or speleology [24,25]. In recent years, TLS has increasingly been used in geomorphological studies to capture more detailed features of given objects and phenomena, such as riverbeds and sediments [26], landslides [27,28,29] or erosion [30,31]. TLS data also allows to derive and illustrate the orientation and slope of rock hillsides [32], respectively destructive phenomena such as falling rocks and avalanches [33,34], debris currents (debris flows) [35] or the boulders instability [36].

With the rapid development of mobile devices, especially smartphones, the computing capacity and data collection through digital photography have been growing. The resolution of the latest smartphone cameras is commonly from 48 to 108 MPix and increasing camera sensor size allows image capturing even in the dark conditions. Three-dimensional point cloud creation from SfM is included in number of photogrammetric applications based on image processing in smartphones [2,37]. Images and processed 3D models are thus useful for mapping of smaller objects such as rock formations, roads or individual tree trunks in forest stands. One of SfM app, commonly used for scientific research, is SCANN3D [2,37]. SCANN3D locally performs the 3D reconstruction, not on the cloud, in quite a rapid way, working entirely offline. The main benefit of the application is the real-time guidance during the image capturing phase, tracking multiple points in the viewpoints, which go from red to green when the camera has been moved sufficiently, ensuring a good image overlap. However, this application does not lock the focus and exposure after acquiring the first image, and this aspect can negatively affect the 3D reconstruction [2].

The Google Tango is technology, that enables to measure and create the models of objects directly using a smartphone. It uses infrared cameras (the RGB-D sensor) to measure image depth. The technology combined three basic principles: image depth estimation based on RGB-D sensor, motion mapping based on inertial navigation (accelerometers and gyroscopes) and so-called SLAM technology [38,39], which can refine the position based on recognized objects in space. The technology has been successfully tested for forest inventory [40,41]. However, Tango was not fully utilized by the market, as the sensor had a high battery consumption and was redundant for the common user. Since 2017, Google has stopped supporting the Tango project and has started offering ARCore technology, which partially replaces it.

ARCore is Google platform for augmented reality experiences building. Using different Application Programming Interfaces (API), ARCore enables a smartphone to sense its environment, understand the world and interact with information. Some of the APIs are available across Android and iOS to enable shared ARCore experiences. ARCore uses three key capabilities to integrate virtual content with the real world as seen through phone’s camera [42]:Motion tracking allows the phone to understand and track its position relative to the world;Environmental understanding allows the phone to detect the size and location of all type of surfaces: horizontal, vertical and angled like the ground, coffee table or walls;Light estimation allows the phone to estimate the environment current lighting conditions.

ARCore’s motion tracking relies on both visual information from the camera and inertial measurements from the device’s inertial measurement unit (IMU). ARCore uses a process called concurrent odometry and mapping, or COM, to understand where the phone is relative to the world around it. ARCore detects visually distinct features in the captured camera image called feature points and uses these points to compute its change in location. The visual information is combined with inertial measurements from the device’s IMU to estimate the pose (position and orientation) of the camera relative to the world over time [43]. 

Although majority of new mobile devices support this technology, there are not too many applications on the market that allow direct mapping of larger objects. Available applications focus only on smaller objects scanning, such as statues or furniture or building interiors, except 3D Scanner for the ARCore app. ARCore’s 3D Scanner uses code from the Tango project library, but it replaces pixel depth calculation with ArCore technology. The 3D Scanner for ARCore uses a sequence of images, which is connected using the SfM algorithm [44] and emulating key points from ARCore. Thanks to the emulation, any mobile phone whose camera supports ARCore technology can be used to scan an object. After the scan is completed, the models are filled using the Poisson reconstruction and texturing using the Tango library. The application is primarily intended for scanning flats and interiors. Prerequisite for good quality results is to dispose of an object with quite complicated surface texture. The main advantage of the application is the reconstruction of objects in real time with very short modeling time depending on the mobile phone performance. Newer application versions also support Time-of-Flight cameras (ToF), which measure the depth of each pixel based on the time of radiation and radiation reception, and primarily serve for accurate focusing [45]. 

The possibility of reducing the costs of the survey with low-cost equipment, such as digital and amateur cameras, smartphones, tablets and action cams, is one of the main advantages of the photogrammetric technique, which also allows for a high repeatability of the survey [2]. Therefore, the main aim of this research is to compare different remote sensing methods for 3D modeling of small rock outcrops on the example of two boulders and to answer the question whether modern technologies of 3D modeling using modern smartphones can offer a simpler and cheaper method for 3D reconstructions of small rock outcrops and other similar objects in other applications, such as archeology. The results of the study should answer whether smartphone applications can be an alternative to conventional data collection methods such as TLS and close-range photogrammetry with sufficient accuracy without the need of expensive instruments (TLS) or professional SfM software.

## 2. Materials and Methods

### 2.1. Study Case

Rock formations in the vicinity of Přečkov are one of the most visited locations in the Třebíčsko Nature Park (Figure 1). They have the character of boulders that are not firmly anchored in the geological bedrock, as it makes them less resistant to natural and anthropic dynamic processes, resulting in rather significant morphological changes. Within a few decades, they are often perceptibly shifted, damaged or destroyed. Three-Dimensional Scanning Technology is a unique method in this case of rock outcrops that greatly supports the observation of relatively rapid changes in their macro and micro shapes, stability and changes in the overall distribution. This monitoring is important for the necessary inventory of this area and for the geodiversity protection and preservation in this part of the Třebíč Nature Park as a valuable natural asset, which has been primarily endangered by anthropic activity. In the past, the rock outcrops were used as an accessible building material and for the colostrum production by burning a pitched pine wood. To do this, it was necessary to modify them by hollowing out the bowl and then to keep the fire and grooves out of the drain. At present, rock outcrops are a very popular place for bouldering—rock climbing without protection at low altitudes, as the surface microstructure is damaged by this activity [46].

In order to compare remote sensing methods, it was necessary to select suitable and easily accessible rock outcrops of a size that would be easy to capture using a smartphone (Figure 2). The selected boulder No. 1 lies on a small hill and is surrounded by pine forest (Figure 3). It´s size is approximately 5 × 4 × 3 m and occupies an area of about 20 m^2^. The selected boulder No. 2 is located at the edge of the ridge above the watercourse at the edge of the forest clearing (Figure 3). It has more complex shape with a size of approximately 6 × 5 × 4 m and occupies an area of about 25 m^2^.

### 2.2. Data Acqusition

Rock formations mapping was performed on 5th December 2018. Technologies of TLS by the FARO FOCUS 3D scanner, imaging using the integrated Xiaomi MI 8 smartphone camera for further SfM processing, 3D Scanner for ARCore and SCANN3D apps were used to create the models. For the purpose of subsequent model referencing, 3 control points near the object were laid out and measured in local coordinate system by total station Trimble M3 for each rock outcrop so that they were visible on the models. Wooden stakes of 5 × 5 cm cross-section were used as control points.

#### 2.2.1. Terrestrial Laser Scanning

The Faro Focus 3D static panoramic scanner was used for the measurement (Table 1). Before that process, basic scanning parameters were set. The resolution 1/8 was chosen, which equals a point spacing of 24 mm at distance of 10 m from the scanner with scanning speed of 122,000 points per second. This point density is sufficient to create an accurate 3D model. Scanning was performed from 6 stations at a distance less than 10 m. The total of 6 spherical spheres with diameter of 20 cm were placed around the rock outcrops for subsequent alignment of individual scans. Individual scans were processed in Faro Scene version 5.3 software by Faro Technologies. The following process consisted in combining individual scans into point cloud based on spherical spheres (separately for each object). The mean registration error of the object Boulder No. 1 was 0.0018 m with a deviation of 0.0015 m, the second object (Boulder No. 2) mean error was 0.0011 m with a deviation of 0.0012 m. No point cloud coloring by images was used to speed up the scanning process. The final model was exported in LAS format.

#### 2.2.2. Close-Range Photogrammetry with SfM Processing in AGISOFT PhotoScan Professional

Both objects were captured by smartphone Xiaomi Mi 8 (Table 2) in the automatic camera mode by gradually bypassing in a circle so that the images overlapped about 70% for successful processing into a model. Altogether, 73 images were taken for the boulder No. 1 and 111 images for the boulder No. 2 (Figure 4). Depending on the incoming light, smartphone camera automatically changed some parameters during data acquisition (ISO about 100, shutter from 1/100 to 1/2000). Images were downloaded from the smartphone and uploaded to AGISOFT PhotoScan software. During processing, autocalibration of cameras and highest quality of alignment and dense cloud generation were used. To determine the scale and position in the coordinate system, it was necessary to mark the control points in the images in advance and enter their coordinates. Subsequently, the model was stored in LAS format for further processing.

#### 2.2.3. 3D Scanner for ARCore

The 3D Scanner for ARCore is an app for Android smartphones that uses the same code base for 3D scanning as the app for Tango. The main difference in these two apps is in providing depth data. The Tango version uses an active depth sensor and ARCore uses passive depth sensing [44]. The mapping was performed using the Xiaomi MI8 with a dual camera by bypassing boulders around with continuous data acquisition. The app uses an automatic regime of camera with no options of settings. The advantage of app is the real-time pre-modeling and the visibility of the scanned surface on the phone display. This allows you to monitor scanned locations and fill in gaps in the model. Both created 3D models were exported to OBJ format for further processing.

#### 2.2.4. SCANN3D App

SCANN3D was developed by SmartMobileVision company and is available only for Android. SCANN3D deploys patent pending photogrammetry technology to enable true 3D model capture and reconstruction for smartphones and tablets. The device becomes a standalone tool to turn images into 3D models—all images are processed by and on it. The resulting 3D models can be stored, shared, and edited by third party applications, and can be used in augmented or virtual reality applications [47]. SCANN3D performs the photogrammetry locally, not in the cloud. SCANN3D supports different formats for further processing (e.g., STL, PLY, OBJ). The mapping was again performed using the Xiaomi MI8 with a dual camera by bypassing rock outcrops around with continuous data acquisition and final models were exported to OBJ format. The application uses the automatic camera mode without the ability to set image properties.2.3. Data Processing and Model Comparison

Exported models from all four methods were processed and compared in the Cloud Compare software and then in ArcGIS for Desktop 10.6.1. In the Cloud Compare all models were edited and aligned to the local coordinate system based on measured control points. Coordinates of control points were assigned to positions of wooden stakes identified in the models. Surface area and volume were calculated, cross-sectioning in 3 axes, as well as comparing the distance between surfaces were performed. It was also necessary to remove terrain points from all models manually so that the resulting model contained only objects of rock outcrops.

In the case of TLS data, created point cloud already had its own scale and altitude determined by the built-in barometric sensor (altimeter). This model can be used directly to calculate the size and volume of an object. Since TLS is a proven method and provides very accurate results, the model created from TLS was used as a reference model for comparison with the other methods (Figure 5).

During the image processing in the AGISOFT software, the coordinates of the control points were entered, so the resulting model had its scale again and was immediately usable for further calculations (Figure 6). The pictures show that the dense clouds on the top of the models have gaps that are caused by insufficient photo overlap.

The model created in 3DScanner for ArCore is also usable almost without modification, as it has its own scale due to the smartphone accelerometer and gyroscope information. It was necessary to rotate the model manually, because the coordinate system axes were interchanged (Figure 7).

The SCANN3D model has no scale and was significantly smaller than other models when it was opened in Cloud Compare. For this reason, it was aligned to the local coordinate system based on the control points coordinates (Figure 8). Figure 5 shows model errors that are caused by overexposure of the first model; in the second model, the shaded areas were underexposed as the model could not be calculated in some places.

The Compute 2.5 Volume tool from Cloud Compare software was used to calculate the volume, surface area and maximum height of rock outcrops. This tool calculates an interpolated raster model with an average point height above the reference plane. To compute the volume Cloud Compare sums the contribution of each cell. This contribution is simply the volume of the elementary parallelepiped corresponding to the cell footprint multiplied by the difference in heights (dV = grid step * grid step * difference of height). Only the cells which have a valid height value for both the ‘ground’ and the ‘ceil’ are used for the global volume estimation. Empty cells or cells for which only one height value is defined are simply ignored [48]. Resolution of 1 cm was chosen for the calculation. To create slices of all models for comparison, it was necessary to align the models so that the slices correspond positively to each other. For this purpose, the models were firstly manually aligned based on 3 identical points and then accurately registered using ICP registration (Interactive Closest Point Registration) [49]. The models were not scaled during registration to maintain their original size and shape. After ICP registration, the Cross Section tool was used to create 3 slices for each model with the largest circumference in the X, Y, and Z axes (Figure 9). These slices were then exported to shapefile format and their perimeter and area were calculated in ArcGIS software. Furthermore, model differences were calculated based on the mutual distance of individual points compared to the reference (TLS) model using the Cloud/Cloud Distance tool.

Models were compared using several methods. At first, the calculated volumes and surfaces were compared with the reference models (TLS) and the deviation values were expressed in units and in % compared to the TLS model. In the next step, perimeters and areas of cross sections in 3 axes were compared.

## 3. Results

### 3.1. Boulder No. 1

In the case of boulder No. 1, the Sfm processing method in AGISOFT PhotoScan achieved the best results as it is professional photogrammetry software and good quality results were expected. The smallest deviations from the TLS model were achieved both when comparing the surface, volume and height of the object (−0.3%, 0.6% and 2.7%) as well as when comparing the perimeter and area of the sections (Table 3, Table 4, Table 5 and Table 6). Comparison of the distance between points and the reference model shows the best results in the case of the 3DScanner for ARCore application with an RMSE of 0.045 m (Table 7). However, in all cases the deviations are relatively small, only the model from SCANN3D achieved lower accuracy (Figure 10).

### 3.2. Boulder No. 2

In the case of boulder No. 2, the results are not so obvious. The 3DScanner for ARCore application, which achieves deviations of up to 5%, is the best based on volume and surface model comparisons. Other methods are less accurate due to imaging conditions (excessive sun exposure of the southern part and shade in the north) as the images could not be optimally aligned (Table 8, Table 9, Table 10 and Table 11). Taking the cross sections and the distance of points from the reference model in comparison, SCANN3D, which most accurately captures the shape of the object, works best (Table 12). However, the upper and northern part of the boulder is missing and therefore it is not the best to assess the surface and volume (Figure 11).

### 3.3. Overall Results

The overall results show that all methods have very small deviations from TLS in percentage units in all observed parameters. Comparison of generated models from different methods with the TLS reference model shows that almost all methods achieve a volume difference of up to 6 m^3^ (10% respectively), with the biggest error in SCANN3D (15 m^3^ and 24% respectively). In the case of the surface, a difference of up to 2 m^2^ (6%) was achieved, again with the greatest error in SCANN3D application (almost 6 m^2^ and 19%). At maximum height, errors are even less than 20 cm (7%). Comparison of cross sections circumference resulted in errors up to 2 m and 10%, in case of cross section area similarly about 2 m^2^ and 10%. A deviation of up to 13 cm was found in all models when comparing the distance between surfaces.

## 4. Discussion and Conclusions

The results of comparison of different remote sensing methods for 3D modeling of small rock outcrops are not entirely clear, and it is not possible to state which mapping method is optimal. Nevertheless, it can be stated that all methods are usable for the purpose of creating rock outcrop models. These surprisingly good results in all image-based methods used are mainly due to the texture of the rocks, as the high texture ensures finding of a high number of tie points. The accuracy of a 3D model depends on the amount of feature points extracted from the captured set of images. If a surface being captured is texture rich, the quality of the produced data would be higher in comparison with poorly textured surfaces. The problem of feature extraction from a texture-less surface is mentioned in many previous studies [50,51,52]. The limiting factors are mainly light conditions during imaging because all methods are passive and are based on the visible part of the electromagnetic spectrum. Good quality model processing is also dependent on the overlay of images, which puts demands during the manual imaging process. In the case of ready-to-use apps like SCANN3D and 3D Scanner for ARCore is process driven by app itself. Compared to TLS, they do not require expensive instruments and achieve very similar results. However, when processing photos in SCANN3D App and AGISOFT PhotoScan, the disadvantage is the necessity to measure at least 3 control points to scale and position in the coordinate system. The 3DScanner for ARCore is the only application that works completely independently, allowing to create a scale model that is directly applicable for further use. Another advantage is the creation of the model in real time, when it is possible to check calculated model during the scanning and fill in the places with the gaps. Ready-to-use applications have limitations primarily in terms of the size of the subject. In the case of rock formations with detailed texture, it is possible to create models of different sized objects, especially depending on accessibility, as these applications require shorter distance to capture feature points. Based on the results presented here, smartphone apps should be restricted to objects with maximum height of 5 m, but the particular constraint is due to the object’s accessibility rather than its size.

Although there is number of 3D scanning applications for smartphones on the market, there are not many studies dealing with the accuracy of the created models of large objects. Similar studies include e.g., [50], where authors compared cloud processing of images taken with different types of smartphones for the purposes of archeological documentation of buildings with the conclusion that the models created are accurate enough, but there are errors at the edges of the model up to 50 cm. Another study [51] focused on reliability of smartphones that incorporate 3D depth sensors for 3D reconstruction of cultural heritage objects. The authors used Google Tango technology and compared results with photogrammetry. The results are very similar as [53], showing the larger deviations on the edges of the model where no perfect match was achieved. The model produced by photogrammetry was of higher quality than the Tango-based one. In [54] the authors concluded that it is possible to obtain high-quality results from numerous images of building captured by smartphone cameras using appropriate software solutions. Very similar results were gained in [55], comparison of riverbank models from TLS and close-range photogrammetry resulting in RMSE from 0,02–0,05 m, depending on the device used (digital single-lens camera x smartphone). In [2] authors tried to create model of the interior of cave by SCANN3D app with conclusion, that due to the higher number of images and to the particularly poor light conditions, the application was able to reconstruct the models only partially, with a quality that was not excellent in terms of the number of points and completeness of the model, even when setting the highest quality level.

Due to the rapid development, in near future, it will be possible to count on further improvements in the 3D data collection using smartphones. In many cases, new smartphones for 2020 already have another (mostly third or fourth) camera for determining the depth of images (ToF cameras—Time of Flight), which should further refine the possibility of creating 3D models according to manufacturer´s information. In [56] the authors tested the ToF camera system with different objects to check the surface of reconstruction and accuracy evaluation. The results have demonstrated that the proposed technique is feasible for dense 3D measurement applications. In [57] the authors developed a machine vision system consisting of RGB and ToF camera to estimate the size of apples in tree canopies. With this system, they were able to not only measure the size of apple fruits but even to calculate the local coordinates of each fruit. For example, the 3D Scanner for ARCore app also allows the use of ToF camera in the new version. The disadvantage is the lower resolution of ToF cameras. However, phones for the year 2020 can have up to 5 Mpix ToF cameras and can improve the data acquisition. As a result, smartphones can replace expensive instruments such as Terrestrial Laser Scanners as well as photogrammetric cameras and desktop software for photogrammetric processing. 

## 5. Conclusions

In this work, we compared different remote sensing methods for 3D modeling of small rock outcrops on the example of two boulders. Four different methods were used for 3D reconstruction: TLS, desktop image processing with SfM and two smartphone apps SCANN3D and 3D Scanner for ARCore. The model size was compared with the TLS reference model. All methods have proved the possibility of using for this purpose 3D modeling. The 3D scanner for ARCore seems to be the best and easiest method because it does not require additional scale and georeferencing of the model with sufficient accuracy. The results confirmed that smartphones can offer cheaper and simpler method of 3D reconstruction than expensive instruments such as Terrestrial Laser Scanners as well as photogrammetric cameras and desktop software for photogrammetric processing. Future research should address the usage of ToF cameras in smartphone for faster and more accurate 3D modeling. 

## Figures and Tables

**Figure 1 sensors-20-01663-f001:**
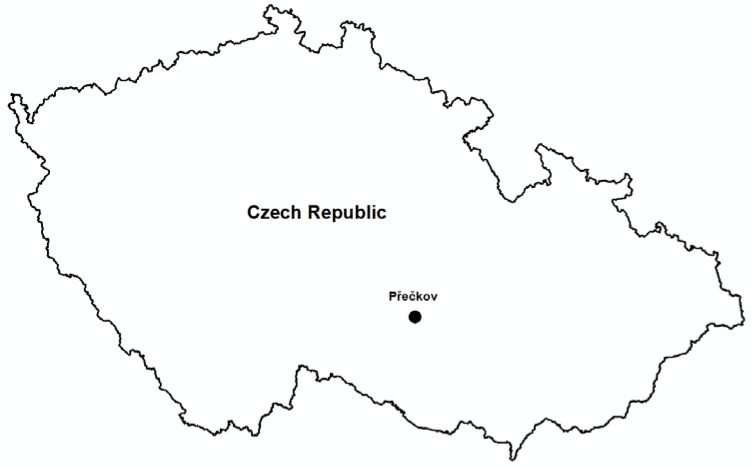
Location of Přečkov within the Czech Republic.

**Figure 2 sensors-20-01663-f002:**
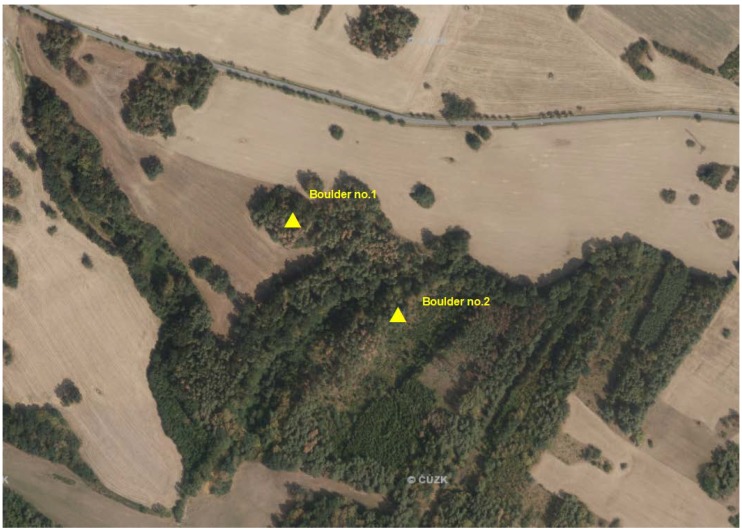
Localization of boulders.

**Figure 3 sensors-20-01663-f003:**
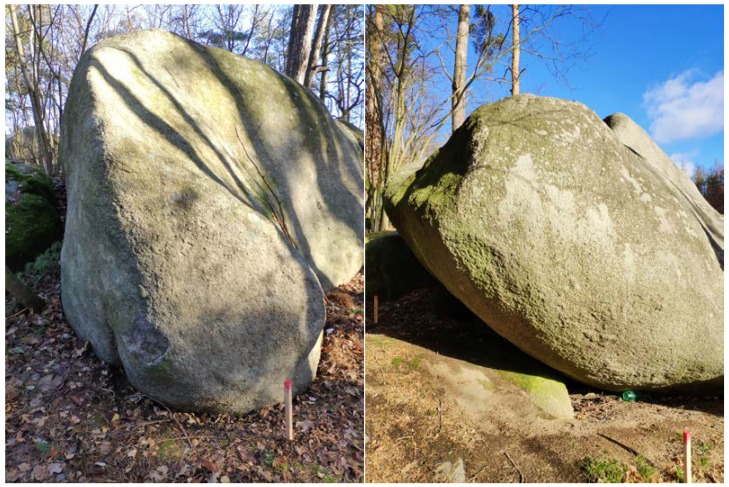
Boulder No. 1 (**left**) and boulder No. 2 (**right**) with visible wooden control points.

**Figure 4 sensors-20-01663-f004:**
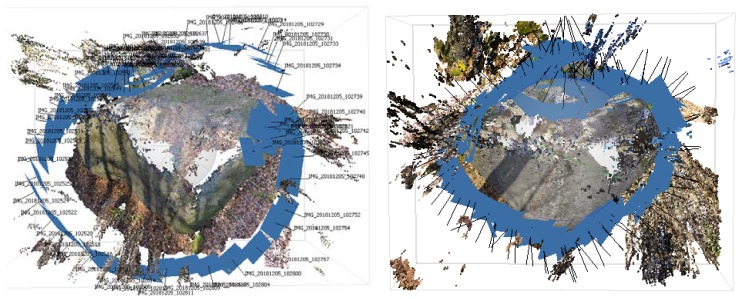
The camera’s selected geometry configuration for data acquisition (Boulder No.1—**left**, boulder No.2—**right**).

**Figure 5 sensors-20-01663-f005:**
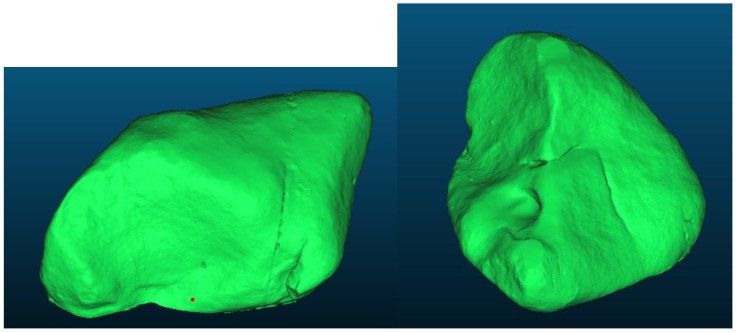
Reference models gained from TLS (Boulder No.1—**left**, boulder No.2—**right**).

**Figure 6 sensors-20-01663-f006:**
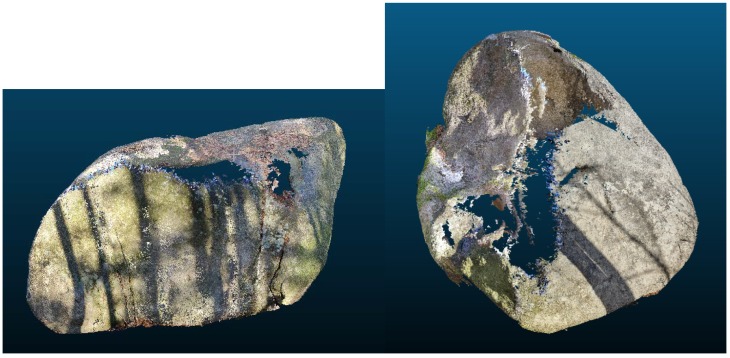
Models by AGISOFT PhotoScan (Boulder No.1—**left**, boulder No.2—**right**).

**Figure 7 sensors-20-01663-f007:**
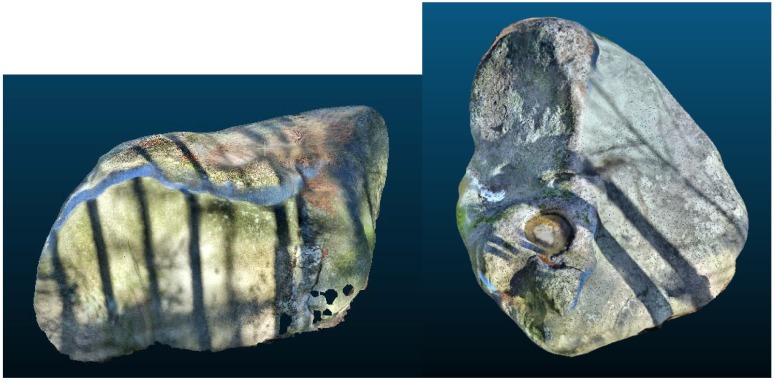
Models created by 3DScanner for ArCore (Boulder No.1—**left**, boulder No.2—**right**).

**Figure 8 sensors-20-01663-f008:**
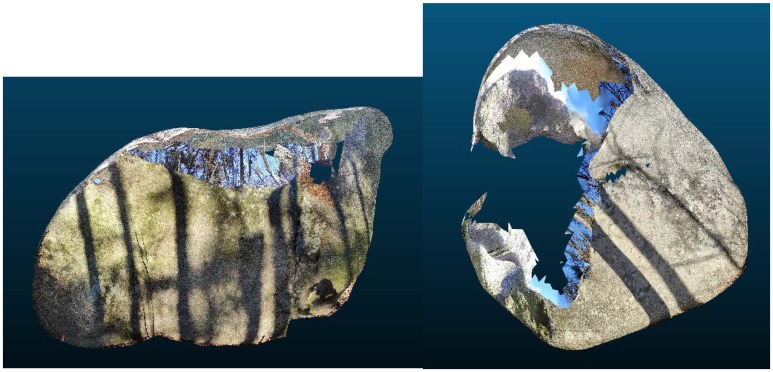
Models from SCANN3D App (Boulder No.1—**left**, boulder No.2—**right**).

**Figure 9 sensors-20-01663-f009:**
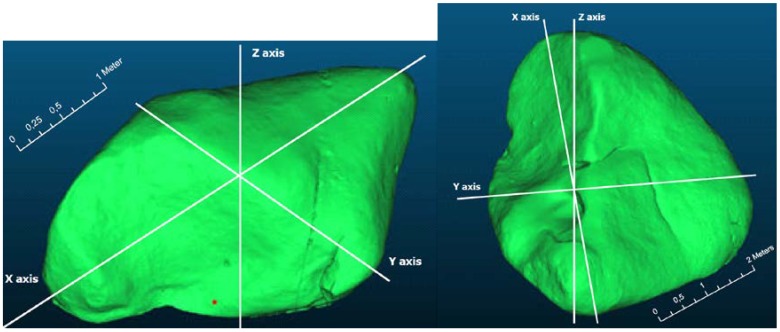
Boulders with marked cross sections in X, Y and Z axes (Boulder No.1—**left**, boulder No.2—**right**).

**Figure 10 sensors-20-01663-f010:**
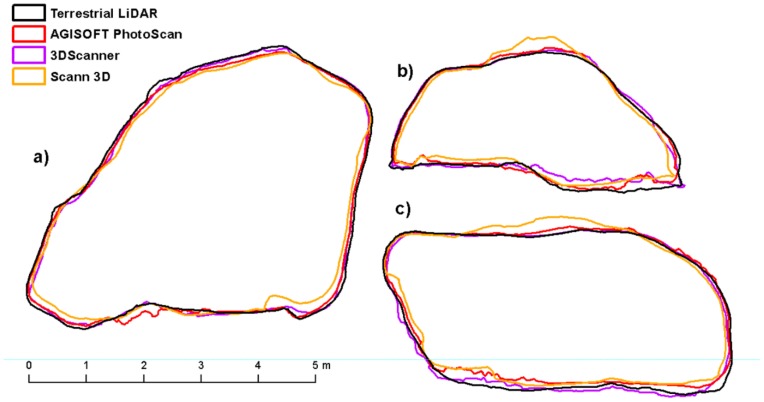
Shapes in cross sections boulder No. 1 (**a**) X axis, (**b**) Y axis, (**c**) Z axis.

**Figure 11 sensors-20-01663-f011:**
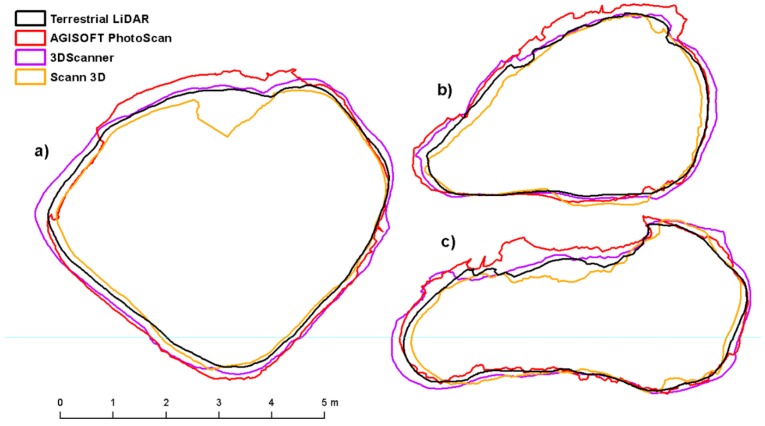
Shapes in cross sections boulder No. 2 (**a**) X axis, (**b**) Y axis, (**c**) Z axis.

**Table 1 sensors-20-01663-t001:** Faro Focus 3D specifications.

Faro Focus 3D Scanner	
**Range:**	0.6–20 m at normal incidence on >10% matte reflective surface, 0.6–120 m indoor or outdoor with low ambient light and normal incidence to a 90% reflective surface
**Measurement speed:**	122,000 / 244,000 / 488,000 / 976,000 points/sec
**Ranging error:**	±2 mm

**Table 2 sensors-20-01663-t002:** Xiaomi Mi2 camera specifications.

Dual Camera	Standard Lens	Telephoto Lens
**Resolution**	12 Mpix	12 Mpix
**Sensor size**	1/2.55″	1/3.4″
**Aperture**	f/1.8	f/2.4
**Pixel size**	1.40 µm	1.00 µm

**Table 3 sensors-20-01663-t003:** Boulder No. 1: volume, surface area and height and particular model differences.

	Volume (m^3^)	Surface (m^2^)	Height (m)	Vol.Diff. (m^3^)	Sur.Diff. (m^2^)	H.Diff. (m)	Vol.Diff. (%)	Sur.Diff (%)	Height.Diff (%)
TLS	45.337	21.906	2.875	ref	ref	ref	ref	ref	ref
AGI	45.033	21.977	2.797	**0.304**	**−0.071**	**0.078**	**0.671**	**−0.324**	**2.713**
3DS	46.536	20.464	3.018	−1.199	1.1442	−0.143	−2.645	6.583	−4.974
S3D	49.752	23.546	3.067	−4.415	−1.640	−0.192	−9.738	−7.487	−6.678

**Table 4 sensors-20-01663-t004:** Boulder No. 1: circumference and the section area.

	Circumference (m)			Area (m^2^)		
	X	Y	Z	X	Y	Z
TLS	15.636	18.252	12.912	14.445	20.680	8.431
AGI	15.937	18.269	12.964	13.849	20.090	8.191
3DS	15.712	18.065	13.276	14.359	20.072	8.032
S3D	15.405	17.437	12.518	13.701	19.096	7.819

**Table 5 sensors-20-01663-t005:** Boulder No. 1: circumference and the section area—model differences.

	Circumference Diff. (m)			Area Diff. (m^2^)		
	X	Y	Z	X	Y	Z
TLS	ref	ref	ref	ref	ref	ref
AGI	−0.301	**−0.017**	**−0.052**	0.597	**0.590**	**0.240**
3DS	**−0.075**	0.118	−0.364	**0.086**	0.608	0.400
S3D	0.232	0.815	0.394	0.744	1.584	0.612

**Table 6 sensors-20-01663-t006:** Boulder No. 1: circumference and the section area—model differences (in percents).

	Circumference Diff. (%)			Area Diff. (%)		
	X	Y	Z	X	Y	Z
TLS	ref	ref	ref	ref	ref	ref
AGI	−1.922	**−0.001**	**−0.404**	4.130	**2.854**	**2.845**
3DS	**−0.482**	0.010	−2.819	**0.595**	2.941	4.739
S3D	1.482	0.045	3.055	5.151	7.662	7.262

**Table 7 sensors-20-01663-t007:** Boulder No. 1: point distances

	Mean (m)	Stand. Dev. (m)	RMSE (m)
TLS	ref	ref	Ref
AGI	**0.028**	0.046	0.054
3DS	0.034	**0.029**	**0.045**
S3D	0.068	0.057	0.089

**Table 8 sensors-20-01663-t008:** Boulder No. 2: volume, surface area and height and particular model differences.

	Volume (m^3^)	Surface (m^2^)	Height (m)	Vol.Diff. (m^3^)	Sur.Diff. (m^2^)	H.Diff. (m)	Vol.Diff. (%)	Sur.Diff (%)	Height.Diff (%)
TLS	64.068	29.018	4.076	ref	ref	ref	ref	ref	ref
AGI	58.057	27.282	3.907	6.011	1.736	0.166	9.382	5.982	4.076
3DS	66.681	27.694	4.178	**−2.613**	**1.324**	**−0.105**	**−4.078**	**4.563**	**−2.578**
S3D	48.720	23.634	3.860	15.348	5.384	0.213	23.956	18.554	5.230

**Table 9 sensors-20-01663-t009:** Boulder No. 2: circumference and the section area

	Circumference (m)			Area (m^2^)		
	X	Y	Z	X	Y	Z
TLS	17.886	13.970	13.445	15.955	23.895	13.027
AGI	18.647	14.999	14.933	16.423	26.295	14.700
3DS	18.805	14.657	15.072	16.951	26.299	14.206
S3D	17.808	13.782	12.598	15.630	22.103	12.504

**Table 10 sensors-20-01663-t010:** Boulder No. 2: circumference and the section area—particular model differences.

	Circumference Diff. (m)			Area Diff. (m^2^)		
	X	Y	Z	X	Y	Z
TLS	ref	ref	ref	ref	ref	ref
AGI	−0.757	−1.029	−1.483	−0.473	−2.395	−1.670
3DS	−0.915	−0.687	−1.622	−1.001	−2.399	−1.176
S3D	**0.082**	**0.188**	**0.852**	**0.320**	**1.797**	**0.526**

**Table 11 sensors-20-01663-t011:** Boulder No. 2: circumference and the section area—particular model differences (in percent).

	Circumference Diff. (%)			Area Diff. (%)		
	X	Y	Z	X	Y	Z
TLS	ref	ref	ref	ref	ref	ref
AGI	−4.232	−7.365	−11.025	−2.968	−10.019	−0.128
3DS	−5.113	−4.916	−12.056	−6.273	−10.039	−0.090
S3D	**0.458**	**1.344**	**6.331**	**2.004**	**7.519**	**0.040**

**Table 12 sensors-20-01663-t012:** Boulder No. 2: point distances.

	Mean (m)	Stand. Dev. (m)	RMSE (m)
TLS	ref	ref	Ref
AGI	0.097	0.085	0.129
3DS	0.081	0.061	0.101
S3D	**0.059**	**0.047**	**0.075**

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
