# Peer review of "Comparison of Different Remote Sensing Methods for 3D Modeling of Small Rock Outcrops"

_sensors, 2020, doi:10.3390/s20061663_

Round 1

Reviewer 1 Report

The paper entlited "Comparison of different remote sensing methods for 3D modelling of small rock outcrops" is an interesting experimental study about the reliability of different 3D methods. However, some minor corrections need to be done:

Abstract:

-Authors should use image-based methods instead of photo-based methods.

Keywords:

-you should avoid the use of acromyns. TLS-->Terrestrial Laser Scanning

Introduction

-line 24-26-->support it with some citations.

-line 41-->

TLS is highly used for preservation analysis. Authors should inlude these citations to support the use of TLS in CH and archeology:

Conde, B., Ramos, L. F., Oliveira, D. V., Riveiro, B., & Solla, M. (2017). Structural assessment of masonry arch bridges by combination of non-destructive testing techniques and three-dimensional numerical modelling: Application to Vilanova bridge. Engineering Structures, 148, 621-638.

Sánchez-Aparicio, L. J., Del Pozo, S., Ramos, L. F., Arce, A., & Fernandes, F. M. (2018). Heritage site preservation with combined radiometric and geometric analysis of TLS data. Automation in Construction, 85, 24-39.

Di Filippo, A., Sánchez-Aparicio, L. J., Barba, S., Martín-Jiménez, J. A., Mora, R., & González Aguilera, D. (2018). Use of a wearable mobile laser system in seamless indoor 3D mapping of a complex historical site. Remote Sensing, 10(12), 1897.

-line 53--> support with new citations

-line 58: technologies instead of echnologies

-line 93: it should recommeded to replace terrestrial stereophotogrammetry by Structure from Motion in order to by in line with the rest of the introduction

Materials and Methods

-Authors should include a new section here. Something like: 2.1-Study Case

-Figure 1: this figure should be supported by an aerial view (or equivalent) of the area studied. It is hard to imaging the enviroment on which the authors focus their research

-Line 119: authors writes about different data-capturing methods. However they dont introduce any table with its technical specifiactions. According to this, it is recommended to introduce a new table (or tables) that deal with the most important specification of each device.

-Line 121: same criteria. Please introduce a table with its technical specifications

-Line 123: It is possible to include in this section (as well as the rests: 2.1,2.2,2.3) new figures?. It is really hard to follow the research. Maybe some images acquired durign the data acquisition and with the devices working?

-Line 129: please clarify the number of scan stations used.

-Line 131: Any image about the data acquisition with the registration spheres?.Which is the diamater of these spheres?

-Line 136: authors write about two scanned objects. Howerver there is not a proper introduction of them. Please write something about the objects scanned in the previous section and support it with some new figures that allow us to imaging the shape of the objects scanned.+

-Line 138: authors should include a plant view with the photogrammetric newtwork used. also you should include in a table the parameters used (e.g. ISO, aperture etc.). It was used an autocalibration method? or the camera was pre-calibrated?. This question need to be clarified in the manuscript.

-Line 166: authors should clarify the method used to register all the point clouds. 

-Line 199: authors should explain more in depth this algorithm.

-Line 206: authors write about ICP. Here it is required the citation: Besl, P.J. y McKay, N., (1992) “A method for
registration of 3-d shapes”, IEEE Transactions
on Pattern Analysis and Machine Intelligence,
pp. 239–256, vol. 14 nº2. 

Section 4

Any future work? to improve this research work.

Author Response

I have tried to correct all the shortcomings and add the article according to requirements of reviewer.

Reviewer 2 Report

See attached. 

Author Response

(The authors gave the same response as above.)

Reviewer 3 Report

Dear Authors,

Thank you for the interesting contribution. 

It would be worth writing a little more about the parameters and settings of the camera used in the phone and lighting conditions when taking pictures.

Was the HDR function used? This could affect lighting problems. Was a tripod used? What lens was used for photos?

Perhaps manipulation of these parameters would show the real possibilities of the method. 

What are the maximum object sizes that can be scanned using smartphones? Considering their current technical parameters?

Author Response

(The authors gave the same response as above.)

Round 2

Reviewer 2 Report

Overall, the manuscript still requires revision as the ultimate research question remains relatively vague, scientific writing style is inconsistent and comparison to relative literature remains lacking. 

Please address the comments raised. 

Author Response

Dear reviewer, I would like to thank you for valuable comments, because I think that they really helped me to make our article much better and make it worth to publish. I tried to correct all meantioned failures and answer your questions. I add also the number of lines where we added and expanded text.

Question 1: 

The authors have clearly taken on board the suggestion to expand on the introduction, by including information regarding the key concepts of the sensors and 3D surface reconstruction hardware and methodologies.

Answer 1:We tried to expand Introduction little bit more and to make it more understandable. Lines 124-134

Question 2:There seems to be no attempt made at expanding on results, and consequently this section lacks sufficient depth. 

Answer 2: We tried to expand this section and added Overall results section. Lines 336-345

Question 3: Much better. You have summarized your paper clearly. Please use this conclusion as a basis for clearly outlining what the intended aim and objectives were to begin with, as these were never quite clear to the reader, and ubsequently, the text comes across as disjointed in sections. 

Answer 3: We tried to define aims and objectives more clearly through the whole text of article. Lines 124-134, 407-416.

Question 4: Still requires rephrasing – ‘The use of photogrammetry and laser scanning significantly *helps to* conduct to effective rocks investigation both on the *of* surface and underground *rock formations*.’

Answer 4: Completely rephrased. Lines 47-53.

Question 5:Lines 48-49: For sentence ‘…for example for coastal cliffs, bedrock ridges [8] or rock slope modeling and 49 discontinuity mapping [9]. The SfM was also used for discontinuity research [10,11].’, please provide an example of discontinuity mapping in a single sentence.

Answer 5: Expanded based on the sources. Lines 47-53.

Question 6: The authors need to provide a more clear and concise research aim i.e. ‘what is your study trying to achieve?’, based on this information. If the main purpose is to compare 3D surface reconstruction between smartphones and TLS, then be direct in this objective. Offering a cheaper alternative in comparison to what?

Answer 6: We tried to better explain aims in Introduction, Discussion and Conclusion. Lines 124-134, 349-350, 407-416.

Question 7: Lines 139-142: Please be consistent with scientific writing style, words such as ‘about’ can be replaced with approximately. Similar issue with ‘It is a little bit bigger..’ Lines 148-149: What about the rocks size made them a suitable choice? Be specific.

Answer 7: We replaced those words and try to be more specific about the suitability of rocks size. Lines 157-162

Question 8: Then be specific. I.e. instead of ‘spatial distribution’, use ‘point density’ or similar.

Answer 8: We used point density instead of spatial distribution. Line 178.

Question 9: Only 3D Scanner for ARCore is mentioned in introduction. Based on lines 150-151 ‘…imaging using the integrated Xiaomi MI 8 smartphone camera for further SfM processing, 3D Scanner for ARCore and SCANN3D apps were used to create the models. ‘ If these are indeed two separate apps using different technology, then you need to address this more clearly.

Answer 9: We tried to find some more sources about principles of SCANN3D app but codes are not free and are not described anywhere. We tried to expand a little bit more in Introduction based on literature sources. Even in those sources are not principles detailly described. Lines 76-82.

Question 10: True. But since there is discussion of scale and coordinate reference issues, the reader expects to see these issues presented i.e. perhaps along an x, y, z scale with reference to a zero origin, that indicates that these issues have indeed been addressed. Simply displaying the 3D models in an arbitrary space is not very informative. Please fix.

Answer 10: We added one figure of boulders with marked cross sections in X, Y and Z axes and with scale. Figure  9.

Question 11: When providing reference letters in any figure the author needs to refer to them in the figure caption. Please fix.

Answer 11: Fixed. Figure capture 10 and 11.

Question 12: Lines 345-346: I’m unsure what you are trying to convey with this sentence ‘Most of the studies dedicated to address the problem of feature extraction from a texture-less surface [53,54,55].’, please consider rephrasing.

Answer 12: Rephrased. Lines 356-357.

Question 13: Line 360: Consider rephrasing ‘Based on own experience these apps are suitable for objects with maximum height of 5 meters’, to something along lines of ‘based on the results presented here, smartphone apps should be restricted to objects with max height of 5m’, or similar. However, please be clear if this is a methodological constraint. 

Answer 13: Rephrased. Lines 370-372.